# Quorum Quenching Mediated Bacteria Interruption as a Probable Strategy for Drinking Water Treatment against Bacterial Pollution

**DOI:** 10.3390/ijerph17249539

**Published:** 2020-12-20

**Authors:** Jia Liu, Xiaohui Sun, Yuting Ma, Junyi Zhang, Changan Xu, Shufeng Zhou

**Affiliations:** 1College of Chemical Engineering, Huaqiao University, Xiamen 361021, China; jialiu0505@outlook.com (J.L.); uvtma@hotmail.com (Y.M.); junyi@hqu.edu.cn (J.Z.); 2Engineering Research Center of Marine Biological Resources Comprehensive Utilization, Third Institute of Oceanography, Ministry of Natural Resources, Xiamen 361005, China; xuchangan@tio.org.cn

**Keywords:** quorum quenching, bacteria pollution, drinking water treatment, bacteria interruption, *Pseudomonas aeruginosa*

## Abstract

*Pseudomonas aeruginosa* in water lines may cause bacteria pollution indrinking fountains that could affect the quality of potable water, thus posing a risk to public health. A clean and efficient strategy is required for drinking water treatment for food safety. In this study, an AiiA-homologous lactonase was cloned from a deep-sea probiotics *Bacillus velezensis* (DH82 strain), and was heterologously expressed so that the capacity of the enzyme on the N-acyl-L-homoserine lactone (AHL)-degrading, effect of bacterial proliferation, biofilm formation and toxic factors release, and membrane pollution from *P. aeruginosa* could each be investigated to analyze the effect of the enzyme on water treatment. The enzyme effectively degraded the signal molecules of *P. aeruginosa* (C6-HSL and C12-HSL), inhibited early proliferation and biofilm formation, significantly reduced toxic products (pyocyanin and rhamnolipid), and inhibited bacterial fouling on the filter membrane, which prevented the secondary contamination of *P. aeruginosa* in drinking water. The findings demonstrated that the quorum quenching enzyme from probiotics could prevent bacteria pollution and improve potable water quality, and that the enzyme treatment could be used as a probable strategy for drinking water treatment.

## 1. Introduction

As a result of the wide usage of drinking fountains and barreled drinking water, bacterial biofilms in drinking fountains and water lines may cause bacterial pollution that directly affects the quality of potable water (PW), and could pose a health risk to the public, usually through secondary bacterial infection [1]. *Pseudomonas aeruginosa* is a typical opportunistic pathogen that forms biofilms in water lines, one of the most critical factors monitored in PW systems [2]. At present, besides the thorough disinfection of instruments and supplied water, common methods of prevention and treatment against *P. aeruginosa* from biofilms are the usage of chemical drugs, such as antiseptics or antibiotics, which might affect food quality and cause problems of drug resistance [3] as a result of excessive release and irregular usage. Therefore, the disruption of bacterial biofilm formation might become a probable strategy to solve this problem [4], by using the bio-active, pollution-free, non-resistant, and non-toxic enzymes as substitutes. 

Quorum sensing (QS) is a phenomenon that regulates bacterial population behavior by sensing the concentration of the auto-inducers, including spore formation, biofilms, and virulence expression [5]. QS isregulated by the threshold nature of signal initiation, the relative specificity of the signal-receptor, and the cascade nature of the regulatory process. *P. aeruginosa* has a typical hierarchical QS system, of which the LasI-LasR system [6] causes the auto-inducer, N-acyl-L-homoserine lactone (AHL), to regulate the downstream RhlI-RhlR [7] and PqsABCDE-PqsR [8] pathways by influencing the transcription and translation of the related genes [9]. Therefore, interference in the level of AHLs leads to quorum quenching (QQ) and has an effect the resulting QS reactions, such as biofilm forming [10] and the virulence expression [11] of *P. aeruginosa*. To date, research about QQ has been mainly focused on the enzymes that degrade the AHL signaling molecules, including three types of AHL, namely AHL-lactonase, -acylase, and –oxidoreductases [12], among which AHL-degrading enzymes are mostly characterized and isolated from the *Bacillus* genus [13]. Studies have also reported that QQ enzymes are sensitive to temperature and pH [14,15], meaning that they could easily be inactivated by water boiling, and cause no harmful effect in drinking water treatment.

In previous work, *Bacillus velezensis* (strain DH82) was isolated from the underlying sea water of a marine deep subsurface at a depth of 6000 m in the Western Pacific Yap trench, and was found to have a broad-spectrum antibacterial ability to affect biofilm-forming for most Gram-negative food-borne pathogens [16,17], and to have the potential probiotic capacity to be applied to drinking water treatment. To address the QQ capacity of DH82 on antifouling against pathogens, and to identify the functional elements in DH82 on drinking water treatment, an AiiA homologous enzyme, named AiiA_DH82_, with 93.7% consensus sequence to the reported N-acyl homoserine lactone hydrolase (PDB: 3DHB), was cloned from the DH82 strain and expressed in thepET28a vector in *E. coli*. The investigation into the potential functional enzyme was focused on QQ activity against *P. aeruginosa* isolated from the water lines on the activity of AHL degrading, effect on biofilms forming and toxic factors releasing, and effect on the control of membrane pollution, in order to verify its capacity on QQ and its ability to halt *P. aeruginosa* contamination as a bioresource in drinking water treatment.

## 2. Materials and Methods

### 2.1. Bacterial Strains and Reagents

*B. velezensis* (DH82 strain; GenBank: MK203035) was isolated from the sea water samples of the Western Pacific Yap trench at the depth of 6000 m, and was kindly offered by the Third Institute of Oceanography (Xiamen, China). *P. aerugnosa* wild-type strain was isolated from the water lines of drinking fountains in this study. The AHL reporter operon of LuxR-P*_luxI-lacO_*-RFP was provided by Xiamen University (Xiamen, China), and was conducted on pET28a in *E. coli* BL21 to detect the level of AHLs. N-Acyl-L-homoserine lactone hydrolase (PDB: 3DHB) from *B. thuringiensis* (GenBank: AY943832), named AiiA_3DHB_, was used as the positive control to analyze the QQ activity of AiiA_DH82_. All of the bacterial strains were cultured in a Luria–Bertani (LB) medium. 

*E. coli* DH5α and BL21 (DE3) competent cells were purchased from Transgen (Beijing, China). The restriction endonucleases and ligation enzymes were purchased from Takara Biotechnology (Dalian, China). The plasmid miniprep kit (Cat. GMK5999) and gel extraction kit (Cat. D2500-02) were purchased from Promega. N-(β-Ketocaproyl)-DL-homoserine lactone (C6-(L)-HSL, Cat. K3255) and (3-Oxododecanoyl)-L-homoserine lactone (3-oxo-C12-(L)-HSL, Cat. 09139) were purchased from Sigma-Aldrich (St. Louis, MO, USA). 

### 2.2. Genetic Engineering and Bioinformatic Analysis of QQ Enzyme

The sequence of AiiA_DH82_ was amplified by PCR from the genomic DNA of DH82 using the primer aiiA-F (5′-atg acagtaaagaagctt tat ttc gtcc-3′) and aiiA-R (5′-tta tat atattcagggaacactttacatcc cc-3′), and was digested by the restriction enzymes *Nde*I and *Xho*I, and subsequently ligated the multipile cloning sites into pET28a with the T4 ligase (Takara, China). The sequence obtained was analyzed by NCBI-BLAST and the maximum likelihood tree was computed using the Poisson correction method by MEGA7.0 software. The 3D structure of the enzyme was simulated using Swiss-model (https://swissmodel.expasy.org/). The bioinformatic information, including the smart domain (http://smart.embl-heidelberg.de/), was analyzed with TMHMM2.0 (https://services.healthtech.dtu.dk/service. php?Tmhmm-2.0).

### 2.3. Protein Expression and Purification

The expression clone of the QQ enzyme was driven by the T7 promoter, and the His-tag coding sequence on pET28a encoded 6xhistidine to the N-terminal of the target protein. The engineered plasmid was transferred in *E. coli* BL21 for protein expression. Then, 0.4 mM of isopropyl-β-d-thiogalactoside (IPTG) was inoculated to the bacterial culture after 3 h in order to induce the expression of AiiA_DH82_. The bacterial pellets were harvested after 20-hof incubation and resuspendeded using a lysis buffer (300 mM NaCl and 50 mM NaH_2_PO_4_ (pH 7.4)), then washed with an imidazole elution buffer (300 mM NaCl, 200 mM imidazole, and 50 mM NaH_2_PO_4_ (pH 7.4)). High affinity NI-NTA chromatography was used to purify the target protein. The purified protein was further analyzed with SDS-PAGE. 

### 2.4. In Vitro Assessment of AHL-Degrading Activity

First, 100 μL of 800 nmol/L AHL was mixed with 100 μL of 1.5 mg/mL purified enzyme solution for pretreatment by letting it stand at 28 °C for 45 min. The *E. coli* carrying reporter operon was cultured overnight in LB media at 37 °C with shaking at 200 rpm/min, and injected with the pretreated AHL solution at 1:100 (*v*/*v*), and was then incubated at 25 °C and 180 rpm/min for 8 h. The bacterial pellets carrying the reporter operon were centrifuged at 8000× *g* at 4 °C, and resuspended with an equal volume of phosphate buffered saline (PBS; 8 mM Na_2_HPO_4_, 137 mM NaCl, 2 mM NaH_2_PO_4_ (PH = 7.4)). The fluorescence intensity of the bacterial pellets was measured with a fluorescence spectrophotometer at 620 nm (excitation wavelength at 584 nm). The AHL-degrading activity of the QQ enzyme was determined by the relative fluorescence unit per cell, which was calculated by dividing the fluorescence intensity at 620 nm to the optical density of the bacterial culture at 595 nm. Each experiment was repeated in triplicate.

### 2.5. Growth Curve of P. aeruginosa

The overnight culture of *P. aeruginosa* was diluted to OD_600_ value at 0.1, and 10% was inoculated (*v*/*v*) in 40 mL fresh LB broth with the addition of 4 mL of 1.5 mg/mL QQ enzyme; this was compared with 4 mL of sterile water, which was used as the negative control. The bacterial culture was sampled at 1, 2, 4, 6, 8, 10, and 12 h in order to obtain the growth curve.

### 2.6. Microplate Biofilm Assay by Crystal Violet Staining

*P. aeruginosa* cultures were grown statically in 200 μL of biofilm medium (BM; filter sterilized tap water supplemented with 5 mM sodium citrate, 0.5% casamino acids, and 0.5% brain heart infusion broth) at 30 °C in a 96-well microplate for 3 d, with the addition of 1.5 mg/mL QQ enzyme in triplicate. After washing with PBS, 200 μL of 0.1% crystal violet solution was added to each well and left for 15 min at room temperature, after which the crystal violet was removed prior to three washes with PBS; 200 μL of ethanol was then added to each well to dissolve any crystal violet bound to the well and any remaining biofilms. After 15 min at room temperature, the absorbance of the wells was measured at 600 nm using the Tecan Infinite M200 Pro.

### 2.7. Assessment of Virulence Factors from P. aeruginosa

The *P. aeruginosa* overnight cultures were diluted to OD_600_ value at 0.1, and mixed with 1.5 mg/mL QQ enzyme; this was compared with the control, which used the same amount of sterile water in triplicate. This was then incubated at 30 °C for 12 h and centrifuged at 8000× *g*/min for 5 min in order to collect the supernatant containing the released virulence factors (pyocyanin and rhamnolipid) generated by *P. aeruginosa*. 

Then, 0.9 mL supernatant of bacterial culture was mixed through with 0.54 mL chloroform, and was allowed to stand for 5 min to collect the supernatant, which was then mixed with 0.2 mL HCl. Consequently, the absorbance of the upper liquid was measured at 520 nm so as to determine the content of pyocyanin.

Then, 1 mL supernatant of bacterial culture was mixed with 4 mL sulphuric acid–anthrone solution (0.2 g anthrone in 100 mL 85% sulphuric acid), and was incubated in boiling water for 15 min. The content of rhamnolipid was determined by the absorbance at 620 nm.

### 2.8. Effect of Antifouling on Watering Fountain

A simulation system of biofilm contamination was assembled to quantify the biofouling of *P. aeruginosa* on the filter membrane, which consisted of a centrifuge tube and a vacuum suction filtration device connected to a catheter with a flow rate controller. The 0.22 μm Polyvinylidene Fluoride (PVDF) filter membrane, pretreated with ultraviolet ray disinfection for 30 min, was placed and fixed steadilybetween the filter cup and the sand-core funnel. *P. aeruginosa* cultures in BM were separately mixed with 1.5 mg/mL QQ enzyme and the negative control of sterile water, and were then continuously flowed into the filter cup at about 6 drops/min under the control of a flow controller, and then flowed through the PVDF filter membrane under using gravity. For 3 d, the bacteria were intercepted so as to accumulate biofilms on the PVDF filter membrane. 

The membrane flux (g/min) was used as an index of biofilm formation on the filter membrane, and was determined by the gravity of sterile water that flowed through the treated PVDF filter membrane per minute. 

Each filter membrane was washed with PBS to remove the planktonic bacteria, and then stained in 2 mL 0.1% crystal violet in a clean petri dish for 15 min at room temperature. After further three washes with PBS to remove the uncombined crystal violet, 2 mL of 75% ethanol was added to the petri dish to dissolve the crystal violet on the remaining biofilm for 15 min. The absorbance of the crystal violet solution at 600 nm was measured using Tecan Infinite M200 Pro so as to determine the biomass of the biofilm on the fouled filter membrane.

### 2.9. Imaging of Filter Membrane

Each PVDF filter membrane was washed with PBS three times, and then fixed with 2.5% glutaraldehyde solution at 4 °C for 2 h. The fixed samples were successively dehydrated with 25%, 50%, 75%, 95% (*v*/*v*) ethanol for 15 min each, and finally with 100% ethanol for 30 min. The dehydrated samples were immediately transferred to a vacuum oven for drying at 55 °C. The dried membranes under went sputter coating with a gold layer and were imaged with a field emission scanning electron microscope (SEM, SU5000, Hitachi, Japan) at 8 kV.

### 2.10. Statistical Analysis

Statistical significance of variance for the collected data was determined by *t*-tests in Origin 9.0 (Originlab, Northampton, MA, USA). A significant difference was indicated using *, a highly significant difference at *p* < 0.01 was marked as ***, a significant difference at 0.01 < *p* < 0.05 was marked as **, and *p* > 0.05 was considered to be no difference.

## 3. Results

### 3.1. Construction of Expression Clone

In this study, as shown in Figure 1, the sequence of *aiiA*_DH82_ was amplified from the genomic DNA of the DH82 strain, and cloned into the pET28a vector between the multiple cloning site of *Nde*I and *Xho*I. The expression clone was driven by T7 promoters and the His-tag coding sequence on the plasmid encoded 6xhistidine to the N-terminal of the target protein; the positive clone was amplified in *E. coli* DH5α and transferred to *E. coli* BL21 for the protein expression. As shown in Figure 2, the result of the sequence alignment showed a93% identity to that of the *Bacillus cereus* Y2 strain, and was homologous with that of *B. thuringiensis* (3DHB, AY943832.1) and *B. Wiedmanni* (C2PHZ1).

The relationships between the other lactonase family are inferred using the maximum likelihood tree, computed using the Poisson correction method, compared with the reported N-acyl-L-homoserine lactone (AHL)-lactonase in the UniPro database. Evolutionary analyses have been conducted in MEGA7.0.

### 3.2. Bioinformatics Analysis and Expression of AiiA

The bioinfomatics of AiiA_DH82_ were analyzed with the online tool ProtParam (http://www.expasy.org/tools/protparam). According to the results, the encoded AiiA_DH82_ was composed of 270 amino acids, with the isopotential point at pH 5.34. Among the amino acid residues, the contents of glutamic acid (Glu), glycine acid (Gly), and leucine acid (Leu) were relatively high, which exceeded 8% of the total components. The grand average of the hydropathicity (GRAVY) was −0.209, which indicated AiiA_DH82_ as a hydrophilic protein. The predicted results also showed no transmembrane signal peptides on AiiA_DH82_, which indicated that AiiA_DH82_ belonged to a cytoplasmic enzyme. Like other known AHL-lactonase, AiiA_DH82_ contained two zinc binding domains (lactonase B and Metallo-beta-lactonase superfamily), and had lactamase activity. The 3D modeling of the enzyme structure was simulated and is shown in Figure 3A. Under the induction of 0.1 mM IPTG, and incubation at 18 °C for 20 h, the target proteins of engineered AiiA_DH82_ were harvested from bacterial extraction by Ni2^+^ affinity chromatography, as shown in Figure 3B, with a concentration of 1.5 mg/mL.

### 3.3. In Vitro Assessment of Ahls Degrading Capacity

The reporter operon, LuxR-P*_luxI-lacO_*-RFP, was constructed to assess in vitro the level of AHLs at free status. As shown in Figure 4, two typical QS signals of *P. aeruginosa*, C6-HSL or C12-HSL, were used as substrates to assess the degrading capacity of the engineered AiiA_DH82_, using the AiiA_3DHB_ from *B. thuringiensis* as the positive control. The control groups that contained untreated AHL were used as the negative control and were labelled as CK1 and CK2. According to the results, the engineered AiiA_DH82_ presented a similar activity to the positive control on the degrading capacity against both C6-HSL (*p* = 0.00065) and C12-HSL (*p* = 0.021), and was observed to have significant difference to control checks (CKs), which indicated the QQ capacity of AiiA_DH82_ on bacterial interruption against *P. aeruginosa* through the effect of the AHL level.

### 3.4. Effect on Bacterial Interruption against P. Aeruginosa

As shown in Figure 5A, the growth curves of *P. aeruginosa* show no significant difference with or without AiiA_DH82_, however, the slope of the bacterial growth curve with the addition of AiiA_DH82_ during the lag phase in the first 2 h was lower than in the absence of the QQ enzyme, which indicates that the QQ enzyme suppressed the increase of the bacterial biomass under a low cell density, and showed no effect on the bacteria once the cell density increased after the logarithmic phase.

As shown in Figure 5B, the presence of AiiA_DH82_ significantly inhibited biofilms forming *P. aeruginosa* (*p* = 0.0013), while the exogenous C6-HSL and C12-HSL showed no significant difference in the biomass increase of the bacterial biofilms, which indicated that AiiA_DH82_ interrupted the biofilms forming *P. aeruginosa* by degrading the AHLs in the bacterial culture, and verified that the exogenous AHLs would not affect the biomasses increasing once the endogenous AHLs were generated over the threshold value.

The virulence factors (pyocyanin and rhamnolipid) were both observed to be significantly down-regulated with the addition of AiiA_DH82_, shown in Figure 5C,D, respectively, whose *p*-values were 0.0000095 and 0.015, respectively. The results demonstrated that the QQ of AiiA_DH82_ mediated the level of AHLs, leading to an inhibition upon the release of pyocyanin and rhamnolipid, which reduced the pathogenicity and potential risks of *P. aeruginosa*.

### 3.5. Trial Experiment of QQ Enzyme on Filter of Drinking Fountain

The bacterial biofilm formation and permeability of the membrane were tested under the treatment of AiiA_DH82_ against *P. aeruginosa*. As shown in Figure 6, treatment of the QQ enzyme significantly inhibited the fouling of *P. aeruginosa* on the PVDF membrane, and the experimental group treated with a free AiiA_DH82_ solution retained a high membrane permeability, while the membrane without the QQ enzyme had been blocked by the biofilm formed by *P. aeruginosa* after 3 d, and the biofilm was visible to the naked eye in Panel A. The results of the crystal violet staining and membrane flux measurement are presented in Panel B, and the SEM images of the fouling layers are in Panel C also prove the antifouling capacity of AiiA_DH82_. The results demonstrated that QQ enzyme treatment could be used as an effective strategy for antifouling as a water purifier in drinking fountains.

## 4. Discussion

The risk of *P. Aeruginosa* pollution does not just exist in the PW system, but also in a range of types of water, including hospital water, drinking water, and non-carbonated bottled water intended for human consumption, which also require the detection and enumeration of the most probable number (MPN) of *P. aeruginosa* (ISO 16266-2:2018), and would benefit from the performance of membrane filtration for water treatment. As an opportunistic pathogen, the physiological and metabolism activities of *P. aeruginosa* have not been extensively studied, which are mediated by QSs [18], including biofilm formation, antibiotic resistance, and virulence factor expression, which affect pathogenicity and host immunity [10,19]. 

AHL-based QS and QQ have been studied for the application of biological wastewater treatment [20], mainly focusing on controlling the membrane biofouling in a membrane bioreactor (MBR) [21,22] using QQ bacteria [23] or enzymes [24]. The QQ enzyme has been found in different bacterial species, among which, AHL-lactonase has been found indifferent *Bacillus* spp. with a high identity of amino acid homology [13]; however, the application of QQ enzymes is still limited, as the sensitivity of temperature and pH [14,15] on the complex environment in MBR may inactivate the enzymes. 

In this study, the engineered AiiA_DH82_ was cloned from a potential probiotics strain isolated from deep-sea water, which showed no harm to the host according to previous studies. The effects of AiiA_DH82_ treatment, at the early stage of bacterial growth with a low cell density, or at biofilm status, both demonstrated the positive inhibition of bacterial biomass accumulation and virulence factor release, by degrading AHLs to regulate the hierarchical QS system. The results of antifouling in the simulation system also demonstrated the direct interruption of AiiA_DH82_ to *P. aeruginosa* in the bacterial culture, which indicated a feasible application in water treatment. As for the microfiltration process, using such a bio-active, pollution-free, non-resistant, and non-toxic enzyme as a substitute [25,26], rather than bacteria strains [23,27], was obviously a more reasonable strategy for drinking water treatment considering food safety and quality control.

On the other hand, AHLs and their analogs are small molecules that cross the bacterial membrane by osmosis, and bind to the receptors in bacteria and trigger the following quorum sensing by targeting protein phosphorylation [28,29]. N-hexanoyl-l-homoserine lactone (C6-HSL) and N-(3-oxododecanoyl)-l-homoserine lactone (C12-HSL) perform a more consequential and eminent function in the biofilm maturation and virulence factor production in *P. aeruginosa* [30,31]. Although the addition of AHL did not significantly improve biofilm forming and virulence factor release, they did have an internal correlation, which might result from the fact that the accumulation of signal molecules reached the threshold and ignored the external addition. This phenomenon suggests that the universal microorganisms of the AHL-based QS system [5], such as Gram-negative bacteria, could be induced or inhibited interactively, and be affected by the addition of AiiA_DH82_.

Although QS is an ideal target to attenuate bacterial virulence and pathogenicity, the effects that QS inhibitors [32] or QQ enzymes [33] have on the broader microorganism communities or that signal alterations might exert on other outcomes in the environment are still of limited knowledge [34]. Therefore, it is highly desirable to further investigate the virulence roles of QS signals or QQ occuring under the influence of the overall environment, in order to diminish unpredicted impacts on food safety.

In order to solve the safety problem of domestic water supply, water purifiers for drinking fountains have become the last way to guarantee the safety of drinking water against pathogens. However, although the filter element of a water purifier could intercept most pollutants, it can easily form a bacterial biofilm on the filter membrane, which would cause membrane blockage and increase the risk of bacterial infection, thus affecting human health. Therefore, the application of the QQ enzyme could be an effective strategy for drinking water treatment, in order to solve the contamination problems caused by AHL-mediated pathogens such as *P. aeruginosa.*

## 5. Conclusions

In this study, the AiiA homologous enzyme was cloned from a potential probiotic bacteria, *B. velezensis*(DH82 strain), isolated from the 6000-m deep subsurface of the Western Pacific Yap trench, and was heterologous expressed in order to investigate its QQ ability on AHL degrading, bacterial interruption in water contamination, and antifouling on the filter membrane. By intermediating the QS of the pathogen via degrading the AHL, the enzyme of AiiA_DH82_ was observed to provide significant inhibition on early proliferation, biofilm formation, and virulent factor release of *P. aeruginosa* (including pyocyanin and rhamnolipid), therefore inhibiting the bacterial fouling of *P. aeruginosa* on the filters of drinking fountains. The findings indicate that the QQ enzyme of AiiA_DH82_ could be used as an effective approach to prevent and control the microbial contamination of drinking water, and to reduce the risk of infection by the opportunistic pathogen *P. aeruginosa*.

## Figures and Tables

**Figure 1 ijerph-17-09539-f001:**
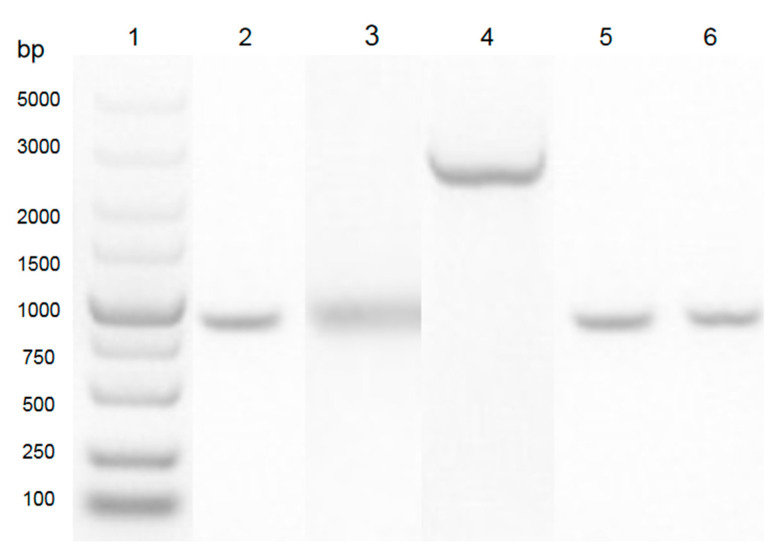
PCR amplification of the *aiiA*_DH82_ gene from *B. velezensis* (DH82 strain). Lane 1: DNA ladder (Takara, 3428Q). Lane 2: PCR product from *B. velezensis* (DH82 strain; *aiiA* homologue gene). Lane 3: DNA fragment of *B. velezensis* (DH82 strain; *aiiA* homologue gene) digested with enzymed *Nde*I and *Xho*I. Lane 4: cloing vector of pET28a digested with enzymes *Nde*I and *Xho*I. Lane 5: positive band confirmed from *E. coli* DH5α. Lane 6: positive band confirmed from *E. coli* BL21.

**Figure 2 ijerph-17-09539-f002:**
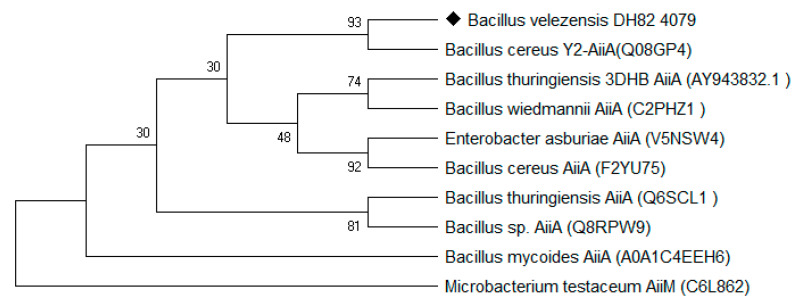
Neighbor-joining tree based on the AiiA from *B. velezensis* (DH82 strain).

**Figure 3 ijerph-17-09539-f003:**
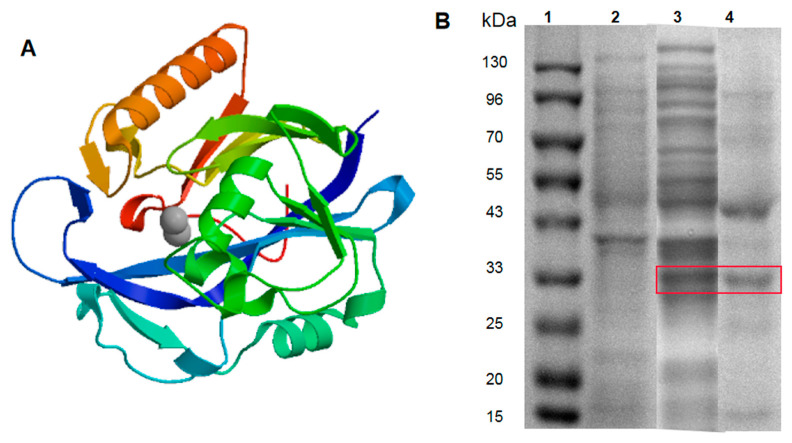
AiiA bioinformatics analysis and protein purification. Panel (**A**): Predicted 3D structure of AiiA_DH82_. Panel (**B**): SDS-PAGE analysis of the AiiA_DH82_ enzyme product. Lane 1: Protein marker. Lane 2: Extraction of non-load pET28a vector. Lane 3: Crude AiiA_DH82_ enzyme extraction. Lane 4: Purified AiiA_DH82_ enzyme extraction. The target protein bands in the gel are shown in the red frame.

**Figure 4 ijerph-17-09539-f004:**
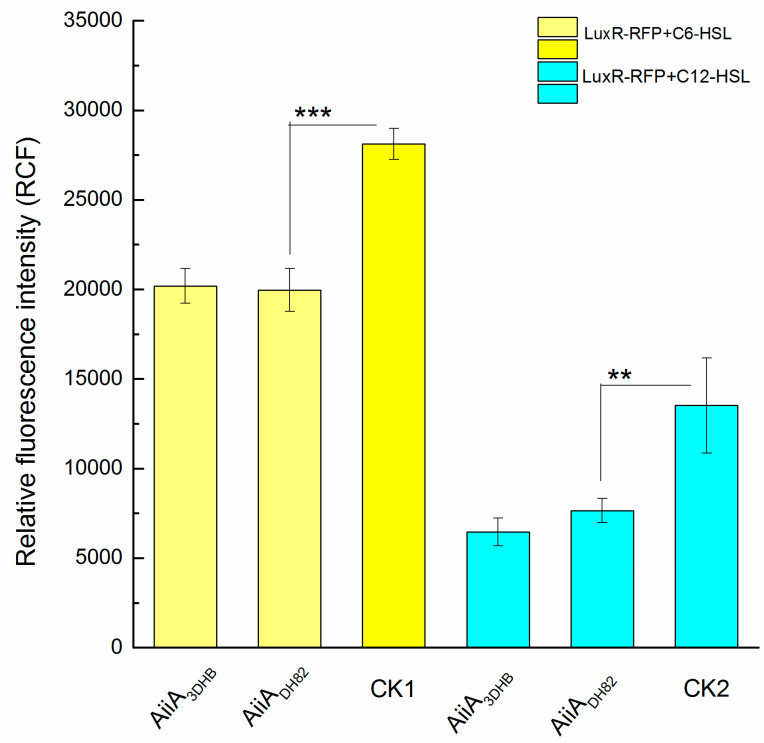
Activities of probable quorum quenching enzymes on AHL degradation. The degradation activities of AiiA_DH82_ aredetermined by the relative fluorescence intensity generated bythe reporter operon (LuxR-P*_luxI-lacO_*-RFP). The experimental groups with added AHL are separately treated with AiiA_DH82_ (C6-HSL in yellow and C12-HSL in cyan), comparedwith the grouptreated with AiiA_3DHB_ (used as the positive control) and the control checks (CKs) of untreated AHL. Error bars are used to determine the standard deviation. Statistical analysis results with a significant difference aremarked using * (significant differences at *p* < 0.01 marked as *** and 0.01 < *p* < 0.05 marked as **).

**Figure 5 ijerph-17-09539-f005:**
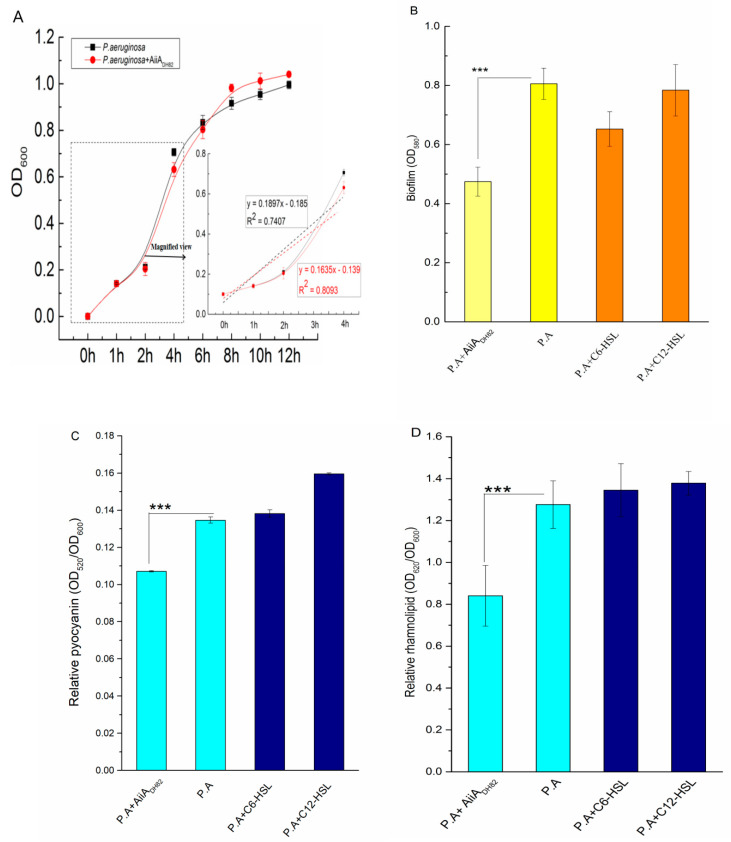
Bacterial interruption of thequorum quenching (QQ) enzyme against *P. aeruginosa*. Here, 1.5 mg/mL AiiA_DH82_ was inoculated to the bacterial culture for treatment, the curve of bacterial density for growth, biofilm accumulation, and amount of released pyocyanin and rhamnolipid were determined by absorbance at 600, 580, 520, and 620 nm, respectively, measured using a microplate reader. Figure 5 shows the (**A**) growth curve of *P. aeruginosa* (bacterial culture with enzyme treatment in red, negative control in black), (**B**) biofilm formed by *P. aeruginosa* (biofilm with enzyme treatment in light yellow, non-treated biofilm in yellow, and biofilm with the addition of AHL in orange), (**C**) released pyocyanin, and (**D**) released rhamnolipid (bacterial culture with enzyme treatment in light cyan, non-treated bacterial culture in cyan, and bacterial culture with the addition of AHLs in navy). Error bars are present to determine the standard deviation. Statistic analysis results with significant difference are marked using * (significant differences at *p* < 0.01 marked as ***).

**Figure 6 ijerph-17-09539-f006:**
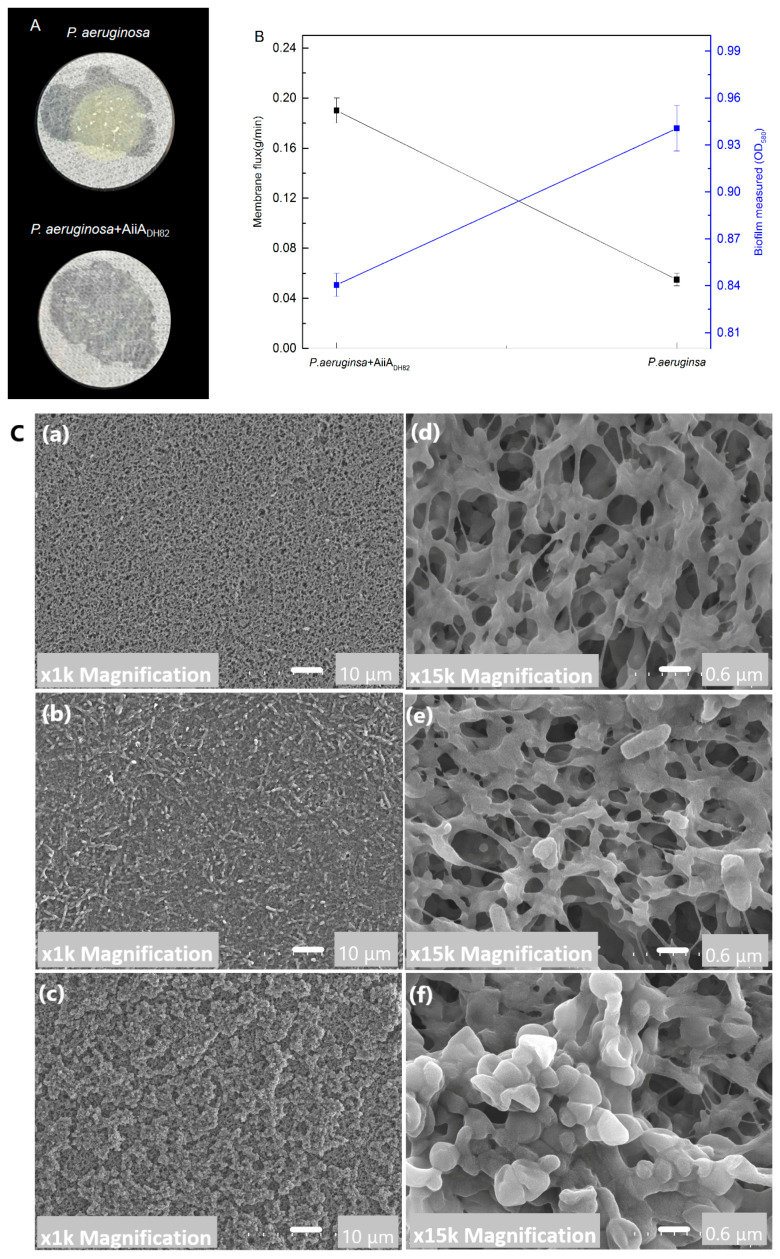
Antifouling capacity of the QQ enzyme against *P. aeruginosa*. Here, 1.5 mg/mL AiiA_DH82_ ismixed with the bacterial culture and is continuously pumped through a 0.22 μm PVDF filter membrane for 3 d. Panel (**A**): Pictures of the filter membrane. Panel (**B**): Biofilm and permeability after treatment. The permeability was determined by the flux of sterile water that flowed through the PVDF treated filter membrane per minute (g/min). The biomass of the biofilm stained by crystal violet is determined by absorbance at 600 nm. The error bars present the standard deviation. Panel (**C**): SEM images of biomass accumulation on the PVDF membrane after 3 d of filtration. (**a**,**d**) New PVDF membrane at 100× and 15,000× magnification; (**b**,**e**) membrane treated with AiiA_DH82_ at 1000× and 15,000× magnification; (**c**,**f**) untreated membrane at 1000× and 15,000× magnification.

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
