# Peer review of "Quorum Quenching Mediated Bacteria Interruption as a Probable Strategy for Drinking Water Treatment against Bacterial Pollution"

_ijerph, 2020, doi:10.3390/ijerph17249539_

Round 1
Reviewer 1 Report
The manuscript present interesting results. Some claims and conclusions regarding potential applications though seem to be an exaggeration. Therefore, mentioning such applications in the title without proper design and experiments with the focus on water quality is unnecessary.
However, the main problem in the manuscript is a serious deficiency in English grammar and style. In the combination with formatting problems, these English problems make the text sometimes barely understandable.
Quite extensive English editing and style correction is needed. Please make also Figure 5A with its expansion clearer.
Author Response
Dear reviewer
Thanks for the kind comments and valuable suggestions.
The purpose of this study is to assess the engineered QQ enzyme on antifouling in filtration and the reduction of violence factors that affect water quality. The design and experimental procedure had been additional described in the revision.
About the English problems, the manuscript had been carefully read through, the errors had been corrected, and the confusing sentences had been revised, including the figure legends, introduction and discussion section.
The Figure 5A had also been replaced with bigger font size and pixel.
Please see the attachment for the revision.
Best regrads

Reviewer 2 Report
Major comments:
1) The introduction contains a lot of information but only 12 references. Quorum sensing in Pseudomonas is very well established and as such the referencing in the introduction needs to be improved.
2) Scanning electron microscopy images are presented but there are no methods to describe what was done. This technique needs to be fully described in its own section in the methods
3) Methods for outlining and quantifying the biofilm deposition on filters needs to be expanded and more information provided
4) Referencing in the discussion is also minimal. This needs to be improved considering many papers have been published on similar research and quorum sensing in Pseudomonas is well characterized and heavily published. Multiple papers have also been published on interfering with quorum sensing system sin Pseudomonas and also on how lactonases can down-regulate virulence factors. None of these papers are discussed. As such, the discussion should incorporate some of these papers and the results obtained obtained from the current study should be compared to them.
==> Based on the above concerns, this reviewer is asking the authors to rewrite and enhance both the introduction and the discussion to cover more relevant literature and prior studies on quorum sensing quenching by lactonases and how this affects virulence factor production.
Minor edits:
Line 23: molecular should be "molecules"
Line 25: rhamnosewith isn't a word. Do the authors mean "rhamnose"?
Line 35: widely should be "wide"
Line 47: you should define QS before using the acronym
Line 62: Expand to both genus and species in the bacterial name
Line 70: "on" should be "in the"
Lines 82-83: B. 83 thuringiensis should be in italics
Line 95: "on" should be "into"
Line 103: authors say the plasmid adds "a histidine", but in actuality the plasmid adds 6 histidines
Line 110: "In vitro" should be in italics
Lines 111-112: Does the purified enzyme solution have a concentration? mg/mL? Volume is irrelevant if concentration is not provided.
Line 112: How was it incubated? standing? shaking? rotating?
Line 114: Please clarify what the "AHLs solution" is? Is it the 800 nmol/L AHLs or the solution mixed with the enzyme? Please include a better description and a concentration.
Line 116: Please clarify the recipe for PBS
Line 119: How was the "relative fluorescence unit per cell" calculated. Were there colony counts done on the culture or was this a comparison of fluorescence to optical density. This needs explanation.
Line 122: "at presence" should be "with the addition"
Line 128" "at the presence" should be "with the addition"
Line 129: "washed" should be "washing"
Lines 153-154: "The biomass of biofilm on each filter membrane was further stained by crystal violet and determined by the absorbance at 600 nm using Tecan Infinite M200 Pro." Provide details on how the biofilm was stained on the filter. This is different than the 96 well plate method which was already described.
Line 162: aiiADH82 should be in italics
Line 163: "on" should be "into the"
Line 165: "a a histidine" should be 6x histidine
Line 172: aiiA should be in italics
LIne 173: aiiA should be in italics
Line 174: "enzymed" should be enzymes
Line 175: replace "checked" with "confirmed" - do this both times
Line 190: domain should be "domains"
Line 191: showed should be "shown"
Line 201: "In vitro" should be in italics
Line 202: "As a small molecule" should be "As small molecules"
Line 202-204: sentence is confusing, consider rewriting
Line 202-204: also add a reference for this statement
Line 204: "in vitro" should be in italics
Line 208: "containing" should be "contain"
Line 209: comparing should be "compared"
Lines 209-211: sentence is confusing, consider revision
Line 213: degrading should be "degradation"
Line 212-219: describe what is in your graph. What do 3DHB and DH82 mean? This is missing from the legend and description? Also state what the error bars represent - are they standard deviation or standard error of the mean?
Line 223: replace "at presence" with "with the addition of"
Line 224: replace "than that at" with "than with the"
Lines 240-247: What is the basis for the color coding in the graphs? This is not described. What are the error bars (standard deviation or standard error of the mean)? In the bar graphs (B, C and D), explain the naming of the different bars in the figure legend.
Lines 262-271: please include magnification for each image in (C) in the figure legend. Current scale bars are very difficult to read even when the image is enhanced.
Lines 262-271: multiple times the authors use "under large scale"or "under small scale" to describe the images in the figure. Please elaborate as these terms are not descriptive and rather vague. Are the authors trying to refer to magnification? if they are, please use quantifiable numbers to refer to magnification (just like above comment)
Line 286: "mediated" should be "mediates"
Author Response
Dear reviewer
We would like to thank you for the kind comments and valuable suggestions, please see the attachment for our revision.
The corrections were also listed below:
Reviewer 2:
Comments and Suggestions for Authors
Major comments:
1) The introduction contains a lot of information but only 12 references. Quorum sensing in Pseudomonas is very well established and as such the referencing in the introduction needs to be improved.
Answer: Thanks for the comments. More references about quorum sensing in Pseudomonas and the strategy of quorum quenching on water treatment had been added in the introduction section.
2) Scanning electron microscopy images are presented but there are no methods to describe what was done. This technique needs to be fully described in its own section in the methods
Answer: Thanks for the suggestions. The procedure of SEM imaging was added in the method section, as described in “2.9 Imaging of fouled filter membrane by scanning electron microscopy”. The details were: “Each PVDF filter membrane was washed with PBS for three times, then fixed with 2.5% glutaraldehyde solution at 4 ℃ for 2 h. The fixed samples were then successively dehydrated with 25%, 50%, 75%, 95% (V/V) ethonol for 15 min each, and finally with 100% ethonol for 30 min. The dehydrated samples were immediately transferred to vacuum oven for drying at 55 ℃. The dried membranes were performed sputter coating with gold layer and imaged with the field emission scanning electron microscope (SU5000, Hitachi) at 8 kV.”
3) Methods for outlining and quantifying the biofilm deposition on filters needs to be expanded and more information provided
Answer: Thanks for the comment and suggestion. The details of experimental procedure to quantify and assess biofilm on filters had been additional described in method section “2.8 Effect of antifouling on watering fountain”.
The description was:
“A simulation system of biofilm contamination was assembled to quantify the biofouling of P. aeruginosa on filter membrane, consisted of a centrifuge tube and a vacuum suction filtration device connected with a catheter with flow rate controller. The 0.22 μm PVDF filter membrane, pretreated with ultraviolet ray disinfection for 30 min, was placed and fixed between the filter cup and the sand-core funnel steadily. P. aeruginosa cultures in BM were respectively mixed with 1.5 mg/mL QQ enzyme and the negative control of sterile water, then continuously flowed into the filter cup at about 6 drops/min under the control of flow controller, and then flowed through PVDF filter membrane under the effect of gravity. The bacteria were intercepted to accumulate biofilm on the PVDF filter membrane for 3 days.
The membrane flux (g/min) was used as an index of biofilm formation on filter membrane, and was determined by the gravity of sterile water that flowed through the treated PVDF filter membrane per minute.
Each filter membrane was washed through with PBS to remove the planktonic bacteria, and then stained in 2 mL 0.1% crystal violet in clean petri dish for 15 min at room temperature. After further three washes with PBS to remove the uncombined crystal violet, 2 mL of 75% ethanol was added to the petri dish to dissolve the crystal violet on the remaining biofilm for 15 min. The absorbance of crystal violet solution at 600 nm was measured using Tecan Infinite M200 Pro to determine the biomass of biofilm on fouled filter membrane.”
4) Referencing in the discussion is also minimal. This needs to be improved considering many papers have been published on similar research and quorum sensing in Pseudomonas is well characterized and heavily published. Multiple papers have also been published on interfering with quorum sensing system sin Pseudomonas and also on how lactonases can down-regulate virulence factors. None of these papers are discussed. As such, the discussion should incorporate some of these papers and the results obtained from the current study should be compared to them.
Answer: Thanks for the comments and suggestion. The discussion section had been revised, using more references about QS and QQ in Pseudomonas and the reported studies on water treatment to discuss the mechanism of quorum sensing in Pseudomonas, and the antifouling strategy by quorum quenching. The hierarchy QS system of P. aeruginosa had also been referenced to describe how did QQ enzyme work on down-regulation of biofilm and virulence factors. The current studies on water treatment had also been discussed in the revision.
==> Based on the above concerns, this reviewer is asking the authors to rewrite and enhance both the introduction and the discussion to cover more relevant literature and prior studies on quorum sensing quenching by lactonases and how this affects virulence factor production.
Answer: Thanks for the comments and suggestions, the introduction and the discussion had been revised with more recent literature on the application of QS and QQ on water treatment, especially on P. aeruginosa. The limitation and application of drinking water treatment had also been mentioned.
Minor edits:
Line 23: molecular should be "molecules"
Answer: The incorrect word “molecular” had been was changed into “molecules”.
Line 25: rhamnosewith isn't a word. Do the authors mean "rhamnose"?
Answer: The incorrect word “rhamnosewith” had been was changed into “rhamnose”.
Line 35: widely should be "wide"
Answer: The incorrect word “widely” had been was changed into “wide”.
Line 47: you should define QS before using the acronym
Answer: The full name of QS had been added.
Line 62: Expand to both genus and species in the bacterial name
Answer: The full name of genus name had been added.
Line 70: "on" should be "in the"
Answer: The incorrect word “on” had been was changed into “in the”.
Lines 82-83: B. 83 thuringiensis should be in italics
Answer: The incorrect font had been was changed to italic.
Line 95: "on" should be "into"
Answer: The incorrect word “on” had been was changed into “into”.
Line 103: authors say the plasmid adds "a histidine", but in actuality the plasmid adds 6 histidines
Answer: The incorrect number had been was changed.
Line 110: "In vitro" should be in italics
Answer: The incorrect font had been was changed to italic.
Lines 111-112: Does the purified enzyme solution have a concentration? mg/mL? Volume is irrelevant if concentration is not provided.
Answer: Thanks for the comment. The concentration of purified enzyme was 1.5 mg/mL, as mentioned in Line 199 in result section. The same purified enzyme sample was used in the experiments as described in materials and methods section. The procedure had been revised as “100 μL of 800 nmol/L AHLs were mixed with 100 μL of 1.5 mg/mL purified enzyme solution for pretreatment by standing at 28℃ for 45 min”.
Line 112: How was it incubated? standing? shaking? rotating?
Answer: The condition of incubation had been detailed as “for pretreatment by standing”.
Line 114: Please clarify what the "AHLs solution" is? Is it the 800 nmol/L AHLs or the solution mixed with the enzyme? Please include a better description and a concentration.
Answer: Thanks for pointing out the mistake. The “AHLs solution” had been described specifically in Line 111-112 by describing the details of pretreatment procedure.
Line 116: Please clarify the recipe for PBS
Answer: The recipe of PBS had been added with “[8mM Na2HPO4, 137mM NaCl, 2mM NaH2PO4, PH=7.4]”.
Line 119: How was the "relative fluorescence unit per cell" calculated. Were there colony counts done on the culture or was this a comparison of fluorescence to optical density. This needs explanation.
Answer: The term of "relative fluorescence unit per cell" had been additional described by “calculated by dividing the fluorescence intensity at 620 nm to the optical density of bacterial culture at 595 nm”
Line 122: "at presence" should be "with the addition"
Answer: The incorrect words had been corrected.
Line 128" "at the presence" should be "with the addition"
Answer: The incorrect words had been corrected.
Line 129: "washed" should be "washing"
Answer: The incorrect words had been corrected.
Lines 153-154: "The biomass of biofilm on each filter membrane was further stained by crystal violet and determined by the absorbance at 600 nm using Tecan Infinite M200 Pro." Provide details on how the biofilm was stained on the filter. This is different than the 96 well plate method which was already described.
Answer: Thanks for pointing out the mistakes. The procedure to assess the biofilm on the filter had been revised as “Each filter membrane was washed through with PBS to remove the planktonic bacteria, and further stained in 2 mL 0.1% crystal violet for 15 min at room temperature in clean petri dish. After tree washes with PBS, 2 mL of ethanol was added to the petri dish to dissolve the crystal violet on the remaining biofilm for 15 min. The biomass of biofilm on fouled filter membrane was determined by the absorbance at 600 nm using Tecan Infinite M200 Pro.”
Line 162: aiiADH82 should be in italics
Answer: The incorrect font had been corrected.
Line 163: "on" should be "into the"
Answer: The incorrect word had been corrected.
Line 165: "a histidine" should be 6x histidine
Answer: The incorrect words had been corrected.
Line 172: aiiA should be in italics
Answer: The incorrect font had been corrected.
LIne 173: aiiA should be in italics
Answer: The incorrect font had been corrected.
Line 174: "enzymed" should be enzymes
Answer: The incorrect word had been corrected.
Line 175: replace "checked" with "confirmed" - do this both times
Answer: The incorrect words had been corrected.
Line 190: domain should be "domains"
Answer: The incorrect words had been corrected.
Line 191: showed should be "shown"
Answer: The incorrect words had been corrected.
Line 201: "In vitro" should be in italics
Answer: The incorrect font had been corrected.
Line 202: "As a small molecule" should be "As small molecules"
Answer: The incorrect words had been corrected.
Line 202-204: sentence is confusing, consider rewriting
Answer: The sentence had been moved to the discussion section, and revised as “AHLs and the analogs are small molecules that cross the bacterial membrane by osmosis, and bind to receptor in bacteria and trigger the following quorum sensing by targeting protein phosphorylation” with cited references.
Line 202-204: also add a reference for this statement
Answer: The references of this statement had been added.
Line 204: "in vitro" should be in italics
Answer: The incorrect font had been corrected.
Line 208: "containing" should be "contain"
Answer: The incorrect word had been corrected.
Line 209: comparing should be "compared"
Answer: The incorrect word had been corrected.
Lines 209-211: sentence is confusing, consider revision
Answer: The sentence had been revised as “According to the compared results, the engineered AiiADH82 present similar activity to the positive control, on the degrading capacity against both C6-HSL (P=0.00065) and C12-HSL (P=0.021), and were observed with significant difference to CKs, which indicated the QQ capacity of AiiADH82 on bacterial interruption against P. aeruginosa by affect the AHLs level.”
Line 213: degrading should be "degradation"
Answer: The incorrect word had been corrected.
Line 212-219: describe what is in your graph. What do 3DHB and DH82 mean? This is missing from the legend and description? Also state what the error bars represent - are they standard deviation or standard error of the mean?
Answer: Thanks for pointing out the mistakes. The “3DHB” and “DH82” respectively means the strain that encodes AiiA, and the labels in the graph had been revised as “AiiA3DHB” and “AiiADH82” to match the description in figure legend. The meaning of error bars had also been additionally described as “Error bars were present to determine the standard deviation”.
Line 223: replace "at presence" with "with the addition of"
Answer: The incorrect words had been corrected.
Line 224: replace "than that at" with "than with the"
Answer: The incorrect words had been corrected.
Lines 240-247: What is the basis for the color coding in the graphs? This is not described. What are the error bars (standard deviation or standard error of the mean)? In the bar graphs (B, C and D), explain the naming of the different bars in the figure legend.
Answer: Thanks for pointing out the mistakes. The color coding of each chats in the graphs had been described as “(A) Growth curve of P. aeruginosa (bacterial culture with enzyme treatment in red, negative control in black); (B) Biofilm formed by P. aeruginosa (biofilm with enzyme treatment in light yellow, non-treated biofilm in yellow, biofilm with addition of AHLs in orange); (C) Released pyocyanin and (D) Released rhamnolipid (bacterial culture with enzyme treatment in light cyan, non-treated bacterial culture in cyan, bacterial culture with addition of AHLs in navy)”. The meaning of error bars had also been additionally described as “Error bars were present to determine the standard deviation”
Lines 262-271: please include magnification for each image in (C) in the figure legend. Current scale bars are very difficult to read even when the image is enhanced.
Answer: Thanks for the comments. The magnification times for each SEM image had been additionally described in the figure legend as “(a and d) new PVDF membrane at 1k times and 15k times magnification; (b and e) membrane treated with AiiADH82 at 1k times and 15k times magnification; (c and f) untreated membrane at 1k times and 15k times magnification”.
Lines 262-271: multiple times the authors use "under large scale"or "under small scale" to describe the images in the figure. Please elaborate as these terms are not descriptive and rather vague. Are the authors trying to refer to magnification? if they are, please use quantifiable numbers to refer to magnification (just like above comment)
Answer: Thanks for the comments. The magnification times for each SEM image had also been labeled in the figures, and the scale had been labeled as well.
Line 286: "mediated" should be "mediates"
Answer: The incorrect tense had been corrected.

Round 2
Reviewer 1 Report
The manuscript quality has been substantially improved. However, English grammar and style still need correction. Figure 5A remains blurry. You can consider to present it as an independent figure of a bigger size.
Please use the correct abbreviation for the journal in Ref. 22 and correct your formatting of the journals in Ref. 27 and Ref. 32 (make all of it italic).
Reviewer 2 Report
The manuscript has been greatly improved with the revisions and this reviewer is satisfied with the revisions regarding the science and specific comments.
The english and grammar will still need further editing so I am recommending english editing to the editor upon acceptance of the manuscript.